# System Design of a Vehicle Based on the Matrix Approach Using Functional Analysis of the Maintenance

**Dušan Mežnar**

IPS, Fakulteta za Organizacijske Vede Kranj, Univerza v Mariboru, 4000 Kranj, Slovenia; dusan.meznar@guest.um.si or dusan.meznar@quest.um.si; Tel.: +386-41-678-862

**Abstract:** The use of extended standard functional analysis of maintenance in the design phase of vehicle structure analysis is presented for the first time, where a matrix of the degree of importance of modules or parts, a matrix size between costs and benefits (costs–benefits), and a logistic support matrix (support index and repair time) are used. The use of these methods allows a designer to be able to determine, in the very early phase of the construction process, the important factors (structure, component price, reliability, repair costs, response time, logistic supportability) that have a major impact on vehicle maintenance. Extended functional analysis also allows us to define critical structures in the project specification of vehicles. A crucial issue in functional analysis is the very extensive implementation of research, drawing conclusions and findings with the basic goal of determining a set of indicators for the verification of assumptions and hypotheses.

**Keywords:** matrix; functional analysis; maintenance; vehicle design; structure design

## 1. Introduction

The main instrument of extended functional analysis (XFA) [1] consists of simple arrays containing a set of main elements, modules, and vehicle assemblies, as well as independent functions. Each cell in the matrix represents a cross-section between the vehicle structure element and the corresponding independent vehicle function.

For XFA, elements, modules, assemblies, and analyzed functions or processes need to be processed according to the hierarchy [2,3]; thus, a list of main components or modules is located on the left side of table, and in the upper part of the matrix is a set of independent vehicle functions. These are determined using the aggregate structure of the vehicule. The advantage of such a deployment is that it allows one to simultaneously monitor and analyze a project and not just when the project is completed; alternatively, a project needs to be completed prematurely and then re-analyzed from the beginning.

The main advantage of extended functional analysis is that a very high accuracy is not needed at the beginning: in fact, it is not necessary to have accurate specifications of parts, data on their properties, price information, or reliability. At this stage, "front to end" analysis is performed [3,4].

In the initial phase of vehicle concept analysis, it is necessary to determine the frame conditions, which are:

1. The basic vehicle is a minibus (basis for this analysis is a minibus produced by an integral bus manufacturer, where the whole vehicle is purposely designed and built for use as a minibus);
2. The implementation is a tourist bus with 29 seats;
3. The highest purchase price of components is approximately Euro 68,000;
4. The vehicle concept will be analyzed based on the following categories:

    4.1. Driving characteristics

    A1—stability;

    A2—speed characteristics;

A3—transience, ascent;
A4—braking;

4.2. Comfort

B1—steerability;
B2—comfort;

4.3. Economics and ecology

C1—noise emission;
C2—fuel consumption.

5. The importance level of functions is evaluated from 1 (lowest priority) to 10 (highest priority).

## 2. Methods

An advantage of the XFA method is that a minimum amount of information is needed; these can be data from a user of the vehicle on its operation, data of vehicle development and testing, or vehicle production and exploitation data. In fact, such an approach is common in the design of a vehicle, as it is independent of the type or purpose of the vehicle. The basic problem encountered is that the data are unstructured, unsystematized, and unreliable. All this, however, has a significant impact on the analyses and their applicability [5].

Checking the applicability of the XFA method in assessing the vehicle concept of system design with respect to vehicle maintenance was performed using a minibus.

Table 1 shows the dataset for the main modules and assemblies of the vehicle, the frequency of failures (*), the reliability of the component (R), and the time spent on average for repairs (MTTR).

**Table 1.** Basic data.

| Main Components | Faults in (%) | Number of Faults (*n*) | Frequency of Faults $\lambda \times (10^{-6})$/h | Reliability (R) | MTTR (h) |
|---|---|---|---|---|---|
| Chassis | 9.0 | 7 | 1.781 | 0.858 | 1.92 |
| Wheels | 1.0 | 2 | 0.400 | 0.967 | 1.51 |
| Engine | 18.0 | 16 | 2.591 | 0.798 | 2.61 |
| Gearbox and clutch | 12.2 | 10 | 2.299 | 0.818 | 3.14 |
| Front axle | 4.0 | 4 | 0.827 | 0.932 | 2.25 |
| Rear axle | 5.0 | 3 | 0.608 | 0.947 | 2.86 |
| Brakes | 17.8 | 12 | 2.299 | 0.818 | 1.41 |
| Steering system | 7.0 | 6 | 1.049 | 0.914 | 2.09 |
| Electronics | 18.0 | 15 | 2.889 | 0.797 | 1.19 |
| Transmission | 8.0 | 5 | 1.049 | 0.914 | 2.51 |
| | 100.0 | 80 | | | |

Data were obtained on a sample group of identical buses (N = 100) over a period of T = 10 years, with the total number of failures being *n* = 80. An exponential breakdown of accident randomness for the vehicle, as well as for key in-vehicle components and systems, was assumed. Figure 1 shows the distribution and frequency of faults more clearly in graphical form.

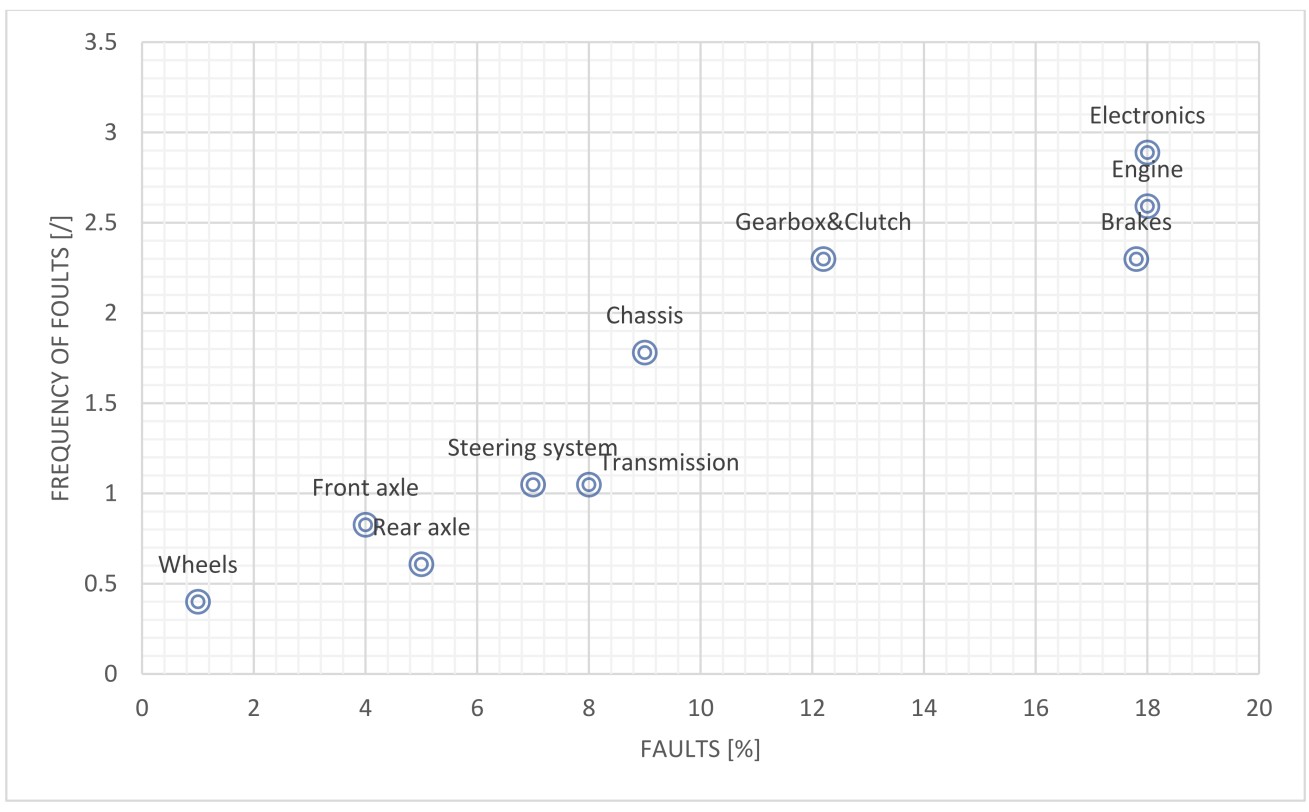

**Figure 1.** Distribution and frequency of faults.

## 3. Formation of the Initial Matrix and the Value Matrix

### 3.1. Basic Assumptions and Value Analysis

The initial matrix (Table 2) is formed by determining the correlations between individual components and the most important functions of the vehicle. Principally, this is a very rough determination of the correlation as greater accuracy would only cause difficulties and thus prevent achieving the purpose of the analysis. Correlations in Matrix 1 are denoted by "x".

The initial matrix gives the designer of the vehicle and logistical support a clear picture of the vehicle structure throughout the design process, first providing a rough idea of the relative importance of each module in a particular function as well as the magnitude of the criticality of each element in the vehicle's structure. Instant information on the selection of function modules and which function is the most complex are also provided.

The motor and the wheels were given the highest level of importance (1) in the structure matrix and the lowest level of the importance (5) was given to the steering system, which, of course, is not necessarily insignificant.

The initial matrix allows to perform parallel analyses of up to 100 correlations between the 10 specific functions of the vehicle and the 10 different elements of the structure. The assessment is made by changing the structure of an item and then pursuing the goal of obtaining the structure of the most important parts (engine, powertrain, electronics, etc.), which meets the criteria regarding the intended characteristics of the vehicle.

The obtained initial matrix provides limited information, but when a relative importance index is introduced for each function from groups A, B, and C, it is possible to extend the functional analysis into the form of a value matrix.

**Table 2.** Initial matrix–structure matrix.

| Main Components | Funct. | A1 | A2 | A3 | A4 | B1 | B2 | C1 | C2 | | Level of Significance |
|---|---|---|---|---|---|---|---|---|---|---|---|
| | | Stab. | Speed | Asc. | Brake | Steer. | Com. | Emis. | Cons. | ΣF | |
| Chassis | | X | - | - | - | X | X | - | - | 3 | 4 |
| Wheels | | X | X | X | X | X | X | X | X | 8 | 1 |
| Engine | | X | X | X | X | X | X | X | X | 8 | 1 |
| Gearbox, clutch | | - | X | X | - | - | - | X | X | 4 | 3 |
| Front axle | | X | - | - | X | X | - | - | - | 3 | 4 |
| Rear axle | | X | X | X | X | - | - | X | X | 6 | 2 |
| Brakes | | X | - | - | X | - | - | - | X | 3 | 4 |
| Steering system | | X | - | - | - | X | - | - | - | 2 | 5 |
| Electronics | | X | - | - | - | X | X | X | X | 5 | 2 |
| Transmission | | - | X | X | - | - | X | - | - | 3 | 4 |
| Components | | 8 | 5 | 5 | 5 | 6 | 5 | 5 | 6 | | |

### 3.2. Structure Matrix: Initial Matrix

The relative value matrix (Table 3) certainly has better information content than that in the initial matrix, and, as such, gives a more informative overview. The sum of the importance index (by rows) gives an index of the relative importance of the components, which in turn allows a designer or analyst to determine the method of delivery (CKD, etc.), suppliers, and the structure. Even though the level of significance is no longer the same, the order for the most important parts remains unchanged.

**Table 3.** Matrix of values.

| Main Components | Function | A1 | A2 | A3 | A4 | B1 | B2 | C1 | C2 | | |
|---|---|---|---|---|---|---|---|---|---|---|---|
| | Index | 7 | 7 | 9 | 8 | 5 | 7 | 5 | 7 | | |
| | | Stab. | Speed | Asc. | Break. | Steer. | Com. | Emis. | Cons. | ΣIP | Lev. of Sign. |
| Chassis | | 7 | - | - | - | 8 | 5 | - | - | 20 | 6 |
| Wheels | | 7 | 7 | 7 | 9 | 8 | 5 | 5 | 7 | 55 | 1 |
| Engine | | 7 | 7 | 7 | 9 | 8 | 5 | 5 | 7 | 55 | 1 |
| Gearbox and clutch | | - | 7 | 7 | - | - | - | 5 | 7 | 26 | 3 |
| Front axle | | 7 | - | - | 9 | 8 | - | - | - | 24 | 4 |
| Rear axle | | 7 | 7 | 7 | 9 | - | - | 5 | 7 | 42 | 2 |
| Brakes | | 7 | - | - | 9 | - | - | 5 | - | 21 | 5 |
| Steering system | | 7 | - | - | - | 8 | - | - | - | 15 | 8 |
| Electronics | | - | 7 | - | - | - | 5 | 5 | - | 17 | 7 |
| Transmission | | - | 7 | 7 | - | - | 5 | - | - | 17 | 7 |

Determining the difficulty factors or the index of importance of singular functions is very subjective as it is based on heuristic methods and experience, but, in the analysis, where the basic goal is a rough estimate, the influence of the indices is not predominant. Interestingly, if all functions in A, B, and C had the same importance indices, this would not affect the change in the level of importance in Matrix 2.

### 3.3. Cost–Benefit Analysis

The XFA method also can yield a cost–benefit analysis matrix, where a comparison between the index of importance of individual functions and relative costs is performed. To perform the comparison, more data are needed; nam ely, in addition to the index of importance of an individual function, the price of all important sets must be determined as well. Cost structure of the main components is presented on Figure 2.

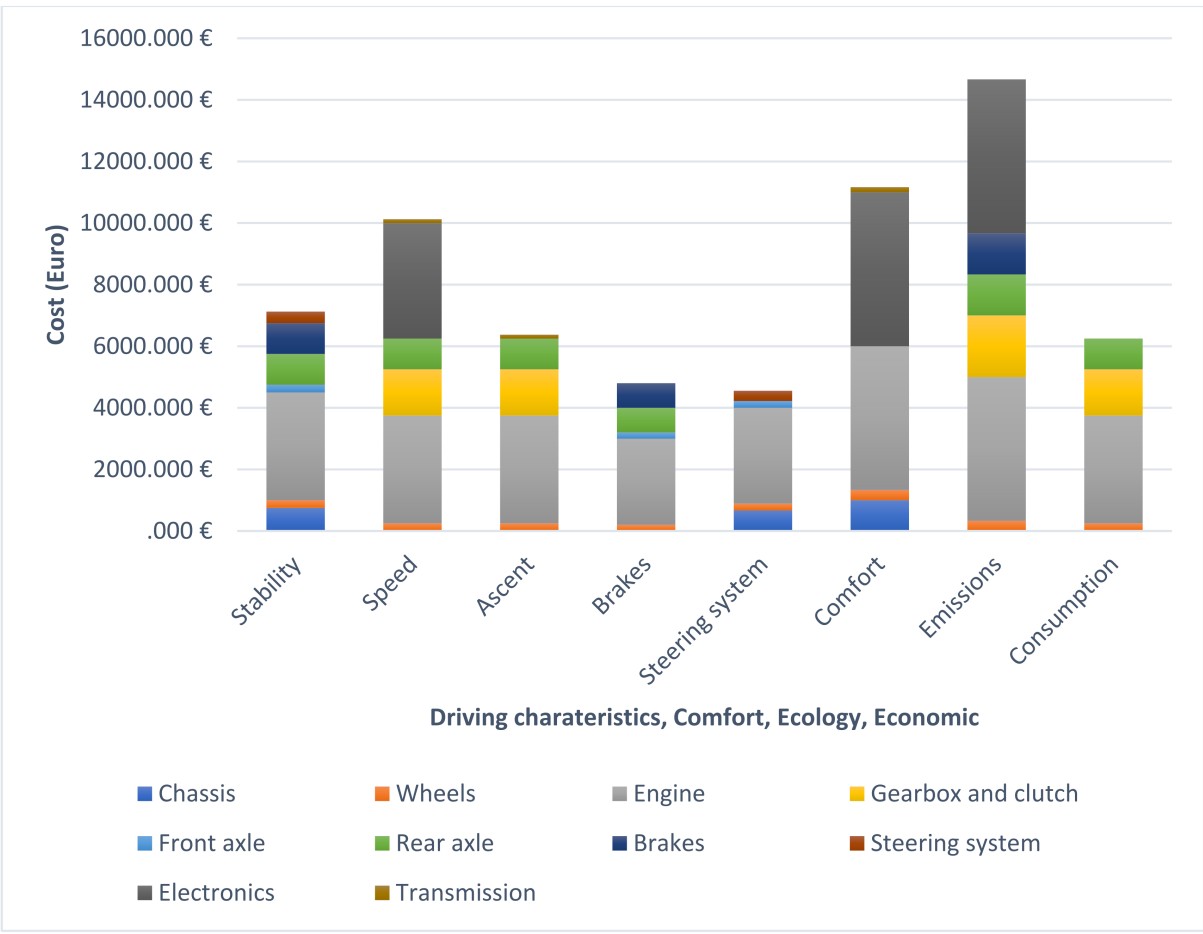

**Figure 2.** Cost structure of the main components relating to the functions of the minibus.

Table 4 shows the prices of the main vehicle modules. This allows to determine a rough estimate of each vehicle function (ΣCOMPONENT line). This also provides a picture of the size of the prices of the individual functions and this gives guidelines for further optimization of vehicle functions.

The so-called cost–benefit ratio, or the ratio between investment (costs) and reimbursed (benefits), is obtained by dividing the value of each set by the significance index; this is so-called cost–benefit ratio.

The column on the right side represents (ΣIP) the repeated values of the indices of importance of sets A, B, and C, and the next column is the cumulative cost–benefit ratio for individual sets. The smaller the obtained value, the greater the benefits of a single module or assembly for a particular vehicle function.

The XFA method enable the traceability of changes in a vehicle's aggregate structure and functions. Tables 1 and 2 show the level of importance of the engine and wheels: 1, electronics—2, etc. Only selected system functions and accompanying assemblies are essential. The cost–benefit analysis matrix (Table 4 and Figure 3), which also contains the prices of sets and the cost–benefit index, shows a significantly different picture. The wheels have the best cost–benefit index (31.7) and level of importance (1), and the motor has a cost–benefit index of 444 and level of importance 9, etc.

**Table 4.** Cost–income analysis matrix.

| Main Components | Function | A1 | A2 | A3 | A4 | B1 | B2 | C1 | C2 | | | |
|---|---|---|---|---|---|---|---|---|---|---|---|---|
| | Index | 8 | 8 | 8 | 10 | 9 | 6 | 6 | 8 | | | |
| | Price | Stab. | Speed | Asc. | Brake. | Steer. | Com. | Emis. | Cons. | ΣIP | Cost–Benefit | Lev. of Sign. |
| Chassis | 4000 | 750 | - | - | - | 667 | 1000 | - | - | 23 | 260.0 | 6 |
| Wheels | 3000 | 250 | 250 | 250 | 200 | 222 | 333 | 333 | 250 | 63 | 31.7 | 1 |
| Engine | 20,000 | 3500 | 3500 | 3500 | 2800 | 3111 | 4667 | 4667 | 3500 | 63 | 444.0 | 9 |
| Gearbox, clutch | 10,000 | - | 1500 | 1500 | - | - | - | 2000 | 1500 | 30 | 400.0 | 8 |
| Front axle | 1000 | 250 | - | - | 200 | 222 | - | - | - | 27 | 74.0 | 3 |
| Rear axle | 6000 | 1000 | 1000 | 1000 | 800 | - | - | 1333 | 1000 | 48 | 166.6 | 4 |
| Brakes | 6000 | 1000 | - | - | 800 | - | - | 1333 | - | 24 | 333.0 | 7 |
| Steering system | 2000 | 375 | - | - | - | 333 | - | - | - | 17 | 176.0 | 5 |
| Electronics | 15,000 | - | 3750 | - | - | - | 5000 | 5000 | - | 20 | 1500.0 | 10 |
| Transmission | 1000 | - | 125 | 125 | - | - | 167 | - | - | 20 | 50.0 | 2 |
| ΣComponents | 80,000 | 5600 | 6480 | 4080 | 38,400 | 32,796 | 53,602 | 70,397 | 40,000 | | | |
| Cost–benefit | | 6125 | 10,125 | 6375 | 4800 | 4555 | 11,167 | 14,666 | 250 | | | |
| Level of imp. | | 3 | 6 | 5 | 2 | 1 | 7 | 8 | 4 | | | |

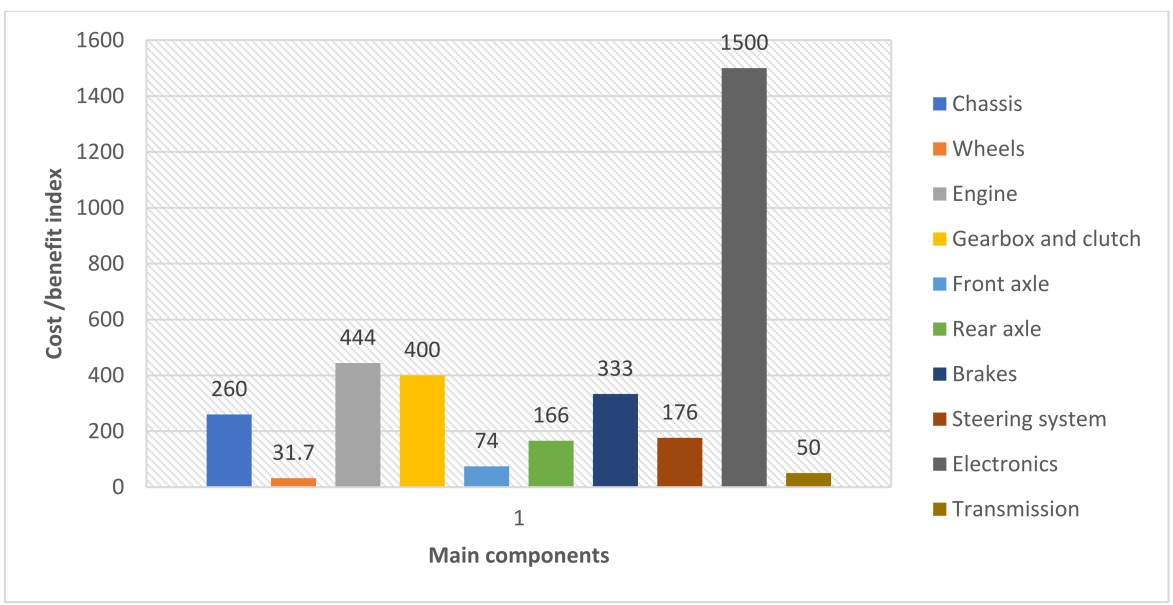

**Figure 3.** Cost–benefit index of the main components.

From the analysis of function forms A2 to C2 in the matrix, we can see that the functions A2—Speed and C1—Emissions are the most expensive. For example, when we try to find ways to reduce production, use, and servicing costs for the C1 emission function, it is relatively easy to find that it has the highest cost–benefit ratio. At the same time, it is also the most expensive feature. This means that it is necessary to make changes in terms of reducing costs and thus improve the cost–benefit index [6,7].

## 4. Expanded Functional Analysis in the Maintenance

XFA, or the method of extended functional analysis, can also be used to assess maintenance of the vehicle. It allows to analyze average time distributions of the corrective maintenance of components and vehicle system MCMH (mean corrective maintenance hours) as shows Table 5.

The basic data of this analysis are the product of the frequency of failures ($\lambda$) and MCMH, and the index of importance of the vehicle function. This provides information on the level of difficulty in maintaining a vehicle. The number of failures per million operating hours (frequency of failures) is a measure of difficulty (see Table 1).

High values of this index are a signal to a designer to optimize, or look for a better solution, in terms of the maintenance and reliability of modules and components in a vehicle. Maintenance difficulty ($\Sigma R$) clarifies the condition of assemblies and modules with respect to maintenance. Matrix 4 shows that the engine and the rear axle have almost the same MCMH, but the maintenance difficulty for the engine is 430—five times higher compared to the ($\Sigma R$) 83 of the rear axles.

The value of maintenance difficulty is also the basis for improvements to the vehicle maintenance concept, relating to the cost of modules and assemblies. An index of costs represents achieved changes and improvements to the concept. In terms of reliability and maintenance concept, priority is given to vehicle assemblies and modules that have the lowest $\Sigma R/C$.

The matrix of value can be also used to analyze the distribution of maintenance support index among vehicle functions. This is obtained through the product during the frequency of failures (time) of individual assemblies and modules relative to price. The higher the value of the index, the more expensive its support.

**Table 5.** MCMH matrix (matrix of mean corrective maintenance hours).

| Main Components | Function | A1 | A2 | A3 | A4 | B1 | B2 | C1 | C2 | | |
|---|---|---|---|---|---|---|---|---|---|---|---|
| | Index | 8 | 8 | 8 | 10 | 9 | 6 | 6 | 8 | | Rentability |
| | (MCMH) $\times$ $\lambda$(10E-6)/h | Stab. | Speed | Asc. | Brake. | Steer. | Com. | Emis. | Cons. | $\Sigma$IP | $\Sigma$R/C |
| Chassis | $1.9 \times 1.78$ | 27 | - | - | - | 30 | 20 | - | - | 77 | 0.01283 |
| Wheels | $1.5 \times 0.39$ | 5 | 5 | 5 | 6 | 6 | 4 | 4 | 5 | 40 | 0.02000 |
| Engine | $2.6 \times 2.59$ | 54 | 54 | 54 | 67 | 67 | 40 | 40 | 54 | 430 | 0.01535 |
| Gearbox, clutch | $3.1 \times 2.30$ | - | 57 | 57 | - | - | - | 43 | 57 | 214 | 0.01783 |
| Front axle | $2.2 \times 0.82$ | 14 | - | - | 18 | 16 | - | - | - | 48 | 0.02400 |
| Rear axle | $2.8 \times 0.61$ | 14 | 14 | 14 | 17 | - | - | 10 | 14 | 83 | 0.01037 |
| Brakes | $1.4 \times 2.30$ | 26 | - | - | 32 | - | - | 26 | - | 84 | 0.01050 |
| Steering system | $2.1 \times 1.05$ | 18 | - | - | - | 20 | - | - | - | 38 | 0.01260 |
| Electronics | $1.2 \times 2.89$ | - | 28 | - | - | - | 21 | 21 | - | 70 | 0.00233 |
| Transmission | $2.5 \times 1.05$ | - | 10 | 10 | - | - | 8 | - | - | 28 | 0.02800 |
| MTTR | | 14.5 | 13.7 | 12.5 | 10.5 | 10.3 | 9.7 | 12.6 | 10 | | |
| Reduced level of functions after a defect | | 116 | 109 | 100 | 105 | 92.7 | 58.2 | 75.6 | 80 | | |
| Level of importance | | 8 | 7 | 5 | 6 | 4 | 1 | 2 | 3 | | |

In the analysis of the support index with respect to all vehicle functions, information on the impact of servicing logistics and spare parts is obtained. This enables significant influence over the design, construction phases, conceptualization of individual vehicle functions and related modules, and assemblies. However, if we consider the index of importance of an individual function, it can assess the criticality of the function in relation with the set goals [8].

Table 6 and Figure 4 show the support index. It is achieved by multiplying the data and represents the sum of the values in each row. Most of the problem's relations to reliability and maintenance are in the engine, electronics, gearbox and clutch, and brakes; this was also expected. In terms of functions, the biggest expected problems relate to speed, comfort, emissions, and braking.

**Table 6.** Maintenance support index matrix.

| Main Components | A1 | A2 | A3 | A4 | B1 | B2 | C1 | C2 | |
|---|---|---|---|---|---|---|---|---|---|
| | 8 | 8 | 8 | 10 | 9 | 6 | 6 | 8 | |
| | Stab. | Speed | Asc. | Brake. | Steer. | Com. | Emis. | Cons. | Index of Cumulative. Support |
| Chassis | 0.01068 | 0.084 | - | - | - | 0.095 | 0.063 | - | 0.242 |
| Wheels | 0.000798 | 0.005 | 0.005 | 0.005 | 0.008 | 0.006 | 0.004 | 0.004 | 0.042 |
| Engine | 0.07252 | 0.579 | 0.579 | 0.579 | 0.720 | 0.649 | 0.434 | 0.434 | 4.560 |
| Gearbox and clutch | 0.02760 | - | 0.219 | 0.219 | - | - | - | 0.164 | 0.825 |
| Front axle | 0.00165 | 0.012 | - | - | 0.016 | 0.014 | - | - | 0.044 |
| Rear axle | 0.00487 | 0.038 | 0.038 | 0.038 | 0.049 | - | - | 0.028 | 0.234 |
| Brakes | 0.0184 | 0.146 | - | - | 0.184 | - | - | - | 0.478 |
| Steering system | 0.00315 | 0.024 | - | - | - | 0.027 | - | - | 0.053 |
| Electronics | 0.0867 | - | 0.693 | - | - | - | 0.519 | 0.519 | 1.734 |
| Transmission | 0.00105 | - | 0.007 | 0.007 | - | - | 0.005 | - | 0.020 |
| **Support** | **0.01068** | **0.888** | **1.541** | **0.848** | **0.977** | **0.791** | **1.025** | **1.149** | **0.245** |

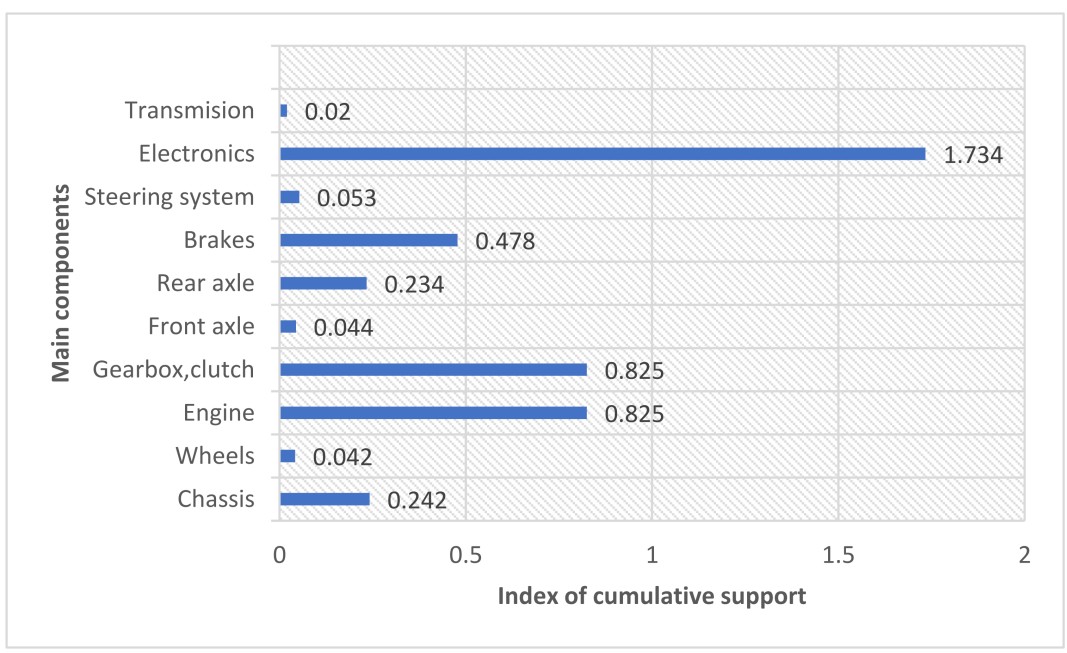

**Figure 4.** Support index for the main components.

## 5. Matrix of Support Index

The obtained value, as a result of analysis of the support index, gives us information about the stock of spare parts and the relationship between the prices of spare parts and

the costs of detecting failures in driving functions and faults with the help of prescribed diagnostic procedures.

The value of the index can be corrected (increased) by taking into consideration the MCMH (time for corrective maintenance) in the calculation process [9,10]. We are not able to process this with the frequency of defects and costs, but it can be done very efficiently with the costs of the assemblies and with the measurement of difficulty in maintenance (MDM) [11].

## 6. Discussion and Results

The main purpose of this paper is to show the use of XFA methods as a very useful tool that allows, at a very early stage of design and conceptualization of a new or modified product, to positively influence decisions and conclusions according to criteria for cost control, maintenance, and design. The interconnectedness of the factors of the analyzed functions provides information on potential problems, which can be solved by making appropriate choices in terms of the structure of modules, sets, or by changing their mutual relations.

The necessary basic data for performing the analysis are assessments of the required aggregate structure of the system, the most accurate price of modules and assemblies, frequency of failures for similar or the same systems, approximate operating conditions, required corrective maintenance time, and importance index of modules and assemblies in a particular system function.

The described XFA method is an excellent basis for strategic decisions, design optimization, cost analysis, or decisions regarding the marketing approach.

In terms of feasibility of future applications and possible challenges, the XFA method needs to be applied to different cases to gain maturity. XFA should be supported by tools relating to experience for simplifying its implementation and to enable to discover information about how a product was made. It will be a great challenge to use and implement XFA in reverse engineering and in the development of new products and services.

In addition, it needs to be supported by tools relating to experience for simplifying its implementation and to enable us to discover information about how a product was made.

In fact, to achieve further development of the XFA method, a validation protocol will be needed. The idea is to structure it with two complementary objectives: first, a validation task with focus on comparative analyses of other methods, and second, a validation task focused on industrial use, which should give answers as to whether this method is applicable in industrial use and give an answer about modeling approaches and tools, which are necessary to improve the XFA method. These issues remain an open perspectives for further work.

**Funding:** This research received no external funding.

**Institutional Review Board Statement:** Not applicable.

**Informed Consent Statement:** Not applicable.

**Data Availability Statement:** Not applicable.

**Conflicts of Interest:** The author declares no conflict of interest.

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
