# Peer review of "System Design of a Vehicle Based on the Matrix Approach Using Functional Analysis of the Maintenance"

_processes, doi:10.3390/pr9050897_

Round 1

Reviewer 1 Report

The author used  XFA methods and  extended  functional analysis in designing and conceptualizing a new or modified product to positively influence decisions with respect to criteria for cost control, maintaining and designing.

The paper is well structured and presented. I have some comments and suggestions:

  1. The paper  has  a gap. It misses to illustrate the data analysis by curves and graphs in order to make effective the obtained data and description.
  2. Line 20, it should be "is consisting".
  3. Line 137, it is "from".

Author Response

Dear Colleague,

Thank you for your revision and valuable comments.

I have corrected the text in line with your comments and the comments of other reviewers, and I hope it now meets your expectations. In line with the requirements of other reviewers, I added some graphs that better illustrate the results to make  the obtained data and description more effective.  In addition, I also described the further challenges and possibilities of further development of the XFA method and its application in the industry.

I did some small linquistic, spell and design corrections as well, but the article structure is remaining the same.

However, I thank you for your revision and discussion, and valuable comments.

Your valuable suggestions gave me new ideas and perspectives for further work. 

Best regards !

Dušan Mežnar

Reviewer 2 Report

  1. This research uses simple matrix analysis combined with vehicle maintenance and cost-benefit analysis to illustrate cases. It is quite advanced in concept and has high practical application value, and is an article of considerable reference value.
  2. The writing of the article is short and concise, and it is also quite easy to understand on the chart description, and the readability is high.
  3. The case in the article is mainly for the analysis of the types of buses. The differences in market functions and price positioning of the various depots are not so great, so the feasibility will be higher. But if it is in the general RV or RV market, because the product design differences between different brands are high, and the definition of the part specifications is inconsistent, how to import the matrix factor will be a big challenge.
  4. As mentioned above, when the composition scale of each factor may be inconsistent, it is necessary to define the "minimum evaluation unit" or "maximum aggregate scale" of the matrix factor.
  5. On the other hand, in the correlation analysis of matrix factors, it is recommended to import three levels of correlation scales (e.g., low correlation, medium correlation, high correlation), so that the weight distribution of the matrix can be refined.
  6. It is quite practical to import functional formulas in functional index and price index, but whether the type of function can only be simple continuous variable polynomial is worthy of further explanation. Since the actual operation site may have many factors of non-quantitative continuous variables (such as categorical scale or nominal scale factors), many studies on multivariate analysis currently use logistic regression for relationship construction and analysis. This part is for the author's reference.
  7. In terms of the feasibility of future applications and possible challenges, it is recommended to discuss more.

Author Response

Dear Colleague,

Thank you for your revision and valuable comments.

I have corrected the text in line with your comments and the comments of other reviewers, and I hope it now meets your expectations. In line with the requirements of other reviewers, I added some graphs that better illustrate the results to make  the obtained data and description more effective. In addition I also described the further challenges and possibilities of further development of the XFA method and its application in industry.

I did some small linguistic, spell, and design corrections as well, but the article structure is remaining the same.

I completely agree with you, the article is a case study of the buses. The basis for this analysis is a minibus produced by an integral bus manufacturer, where the whole vehicle is purposely designed and built for use as a minibus. The approach is quite different compared with the designing and production process of the most basic approach which is the van conversion, where the minibus is derived by modifying the existing van design or bodybuilding principle where a manufacturer builds a specific body for fitting to a semi-completed van or light truck chassis. I am not an expert in RV production, but I assume that the designing and production process is very similar to the van conversion or bodybuilding of the buses. I would like to apply the method to the product specifics of the bus company where I work. Our company is making its own design of chassis and the buses and despite this, I was focused on the main bus characteristics which the designer can influence. In the case of RV, the situation is a little bit different, because they are usually using existing vans or truck platforms, so they can only influence the body and interior.

Regarding the three levels of correlation scales,  I agree with you as well! In the described case of the minibus, we have three different categories (Driving characteristics, Comfort, Economics, and Ecology) with 2-4 elements. Due to my opinion, this case is not so complex that the use of three levels of correlation scales will be needed, but it any case is a good idea! In this case, the focus was on the cost-benefit issue. The level of correlation was indirectly defined by the costs of the components. The basic target of this study is to make use of XFA applicable in the bus designing process and find operational solutions to ensure easy implementation in an industrial context. Once the analysis is done, the functional specifications of the system are translated into technical solutions where only designers start thinking about solutions.

However, I thank you for your revision and discussion, and valuable comments.

Your valuable suggestions gave me new ideas and perspectives for further work. 

Best regards !

Dušan Mežnar

Round 2

Reviewer 1 Report

All suggestions have been considered. I am satisfied with this version....